# Lectin-Modified Magnetic Nano-PLGA for Photodynamic Therapy In Vivo

**DOI:** 10.3390/pharmaceutics15010092

**Published:** 2022-12-27

**Authors:** Vera L. Kovalenko, Elena N. Komedchikova, Anna S. Sogomonyan, Ekaterina D. Tereshina, Olga A. Kolesnikova, Aziz B. Mirkasymov, Anna M. Iureva, Andrei V. Zvyagin, Petr I. Nikitin, Victoria O. Shipunova

**Affiliations:** 1Moscow Institute of Physics and Technology, 9 Institutskiy Per., 141701 Dolgoprudny, Russia; 2Shemyakin-Ovchinnikov Institute of Bioorganic Chemistry, Russian Academy of Sciences, 16/10 Miklukho-Maklaya St., 117997 Moscow, Russia; 3Prokhorov General Physics Institute, Russian Academy of Sciences, 38 Vavilov Street, 119991 Moscow, Russia; 4Nanobiomedicine Division, Sirius University of Science and Technology, 1 Olympic Ave., 354340 Sochi, Russia

**Keywords:** lectin, PLGA, ConA, magnetic polymer nanoparticles, MPQ, allografts, photodynamic therapy, IR775, image-guided therapy

## Abstract

The extreme aggressiveness and lethality of many cancer types appeal to the problem of the development of new-generation treatment strategies based on smart materials with a mechanism of action that differs from standard treatment approaches. The targeted delivery of nanoparticles to specific cancer cell receptors is believed to be such a strategy; however, there are no targeted nano-drugs that have successfully completed clinical trials to date. To meet the challenge, we designed an alternative way to eliminate tumors in vivo. Here, we show for the first time that the targeting of lectin-equipped polymer nanoparticles to the glycosylation profile of cancer cells, followed by photodynamic therapy (PDT), is a promising strategy for the treatment of aggressive tumors. We synthesized polymer nanoparticles loaded with magnetite and a PDT agent, IR775 dye (mPLGA/IR775). The magnetite incorporation into the PLGA particle structure allows for the quantitative tracking of their accumulation in different organs and the performing of magnetic-assisted delivery, while IR775 makes fluorescent in vivo bioimaging as well as light-induced PDT possible, thus realizing the theranostics concept. To equip PLGA nanoparticles with targeting modality, the particles were conjugated with lectins of different origins, and the flow cytometry screening revealed that the most effective candidate for breast cancer cell labeling is ConA, a lectin from *Canavalia ensiformis*. In vivo experiments showed that after i.v. administration, mPLGA/IR775–ConA nanoparticles efficiently accumulated in the allograft tumors under the external magnetic field; produced a bright fluorescent signal for in vivo bioimaging; and led to 100% tumor growth inhibition after the single session of PDT, even for large solid tumors of more than 200 mm^3^ in BALB/c mice. The obtained results indicate that the mPLGA/IR775 nanostructure has great potential to become a highly effective oncotheranostic agent.

## 1. Introduction

Nanobiotechnology opens up new possibilities in the diagnostics and treatment of socially significant diseases, including oncological ones [1,2,3,4]. A number of drugs based on nanoparticles have already been approved for clinical applications [5]. For example, liposomal forms of chemotherapeutic drugs, such as doxorubicin, penetrate the tumor through the EPR effect (enhanced permeability and retention effect) [6]. However, for the majority of human solid tumors, in contrast to laboratory rodents, the EPR effect is much less pronounced, and passive delivery is not as efficient as expected [7].

To solve this problem, different methods of targeted delivery are now being developed, e.g., targeted drug delivery, magnetic delivery, or combination delivery strategies, thus implementing a “magic bullet” concept. Namely, magnetically enhanced nanoparticle delivery is one of the mainstream directions of modern nanobiotechnology. Magnetite incorporation into nanoparticle structure allows for non-invasively visualizing different pathologies within the organism with different MRI regimens [8,9], quantitatively assessing the nanoparticle accumulations in different tissues in fundamental applications [10], and performing a magnetically-guided drug delivery using an external source of electromagnetic field near the region of interest, e.g., the tumor site [11,12]. Different magnetite-based nanoparticles are already used in clinical applications as MRI contrasting agents, thus proving the effectiveness of this kind of nanoparticle for biomedicine (ferucarbotran for hepatocellular carcinoma or ferumoxide for the imaging of mononuclear phagocyte systems) [13].

Another actively developing approach is the targeted delivery strategy of different drugs to the tumor site. To impart cancer-cell-targeting modality to the nanoparticle, the particle surface is equipped with different recognizing molecules. Different proteins are traditionally used for the targeted delivery of nanoparticles to cancer cells: antibodies and their derivatives [14,15]; scaffold polypeptides of various natures, such as DARPins or affibodies [16,17,18,19,20]; transferrin [21]; lactoferrin [22]; and various peptides [23]. However, despite the variety of tools for targeted drug delivery and the number of fundamental works devoted to this topic, there are currently no targeted nanomedications approved for human administration.

This is due both to the difficulty of marketing new compounds and the insufficient efficiency of existing candidate nanoformulations. The described problems require the development of new smart nanosystems with a fundamentally different mechanism of action that selectively affects cancer cells. In particular, we believe that it is necessary to develop nanoformulations that would target the tumor with greater efficiency and in which the anti-cancer agent would have maximum cytotoxicity, but being activated only in the tumor area in order to reduce side effects: hepatotoxicity and cardiotoxicity.

One alternative way to affect cancer cells is to target their glycosylation profile that differs from that of normal ones [24,25,26,27,28,29]. Aberrant glycosylation is one of the significant hallmarks of cancer cells that has contributed to cancer progression, angiogenesis, and metastasis, and therefore is a promising target for drug delivery [30]. In this regard, proteins that specifically bind carbohydrate residues in the composition of cell membrane proteins present a promising alternative to existing targeting molecules [31,32]. In particular, various lectins, including those of plant origin, are able to specifically and reversibly bind carbohydrate residues in other biomolecules. We do believe that lectin-modified nanoparticles combined with magnetically guided delivery present a promising alternative to the existing cell targeting strategies for the implementation of different types of cancer therapy: chemotherapy, photothermal therapy, photodynamic therapy, and so on.

In particular, one of the effective methods of fighting tumors is photodynamic therapy (PDT) based on the conversion of external electromagnetic radiation into reactive oxygen species (ROS) that are harmful to cancer cells. The main advantage of PDT in comparison to, e.g., chemotherapy or surgery, is the full non-invasiveness and the activation only with light at the site of action, thus making undesirable side effects negligible.

We believe the polymer PLGA nanoparticles to be the most effective matrixes for the PDT sensitizers incorporation, since the copolymer of lactic and glycolic acids is fully biocompatible, biodegradable, and makes possible slow release of PDT sensitizer from the nanoparticle at the tumor site. PLGA-based nanoformulations are already approved for human use and confirmed their efficacy for biomedicine [5,33].

Currently, PLGA nanoparticles conjugated with lectins were shown to significantly improve the delivery and cytotoxic efficacy of chemotherapy in vitro and in vivo. Namely, wheat-germ-agglutinin-conjugated PLGA nanoparticles loaded with paclitaxel were found to be more effective at cell cycle arrest of A549 cells compared to free paclitaxel in vitro and had greater tumor doubling time (25 vs. 11 days) in vivo [34]. Despite these promising results, currently, there are only a few studies on the use of lectin-conjugated PLGA nanoparticles for the delivery of chemotherapeutic drugs, and there are no data on the use of such promising lectin-conjugated PLGA nanoparticle formulations as hybrid magnetic-PLGA nanoparticles enabling MRI tracking and nanoparticles for photodynamic drug delivery, which allow for a non-invasive tumor treatment.

Here, we demonstrate the rational design of lectin-modified nano-PLGA loaded with PDT sensitizer (IR775) and magnetite, namely, mPLGA/IR775-lectin. The developed nanoparticles realized a combined targeted delivery strategy, namely, the combination of active targeting mediated by lectin and magnetically guided targeting mediated by magnetite loaded inside nanoparticles and an external magnetic field. The anti-cancer efficacy is mediated by the loading of the photosensitizer, IR775 dye, into PLGA nanoparticle structure. The heptamethine cyanine derivative IR775 is one of the most effective photosensitizers, especially inside polymer nanoparticles, due to its hydrophobic nature; under external light irradiation in the near-infrared window in biological tissue, it produces reactive oxygen species (ROS) leading to cancer cell death, thus affecting only the specific tissue site only on demand under light exposure [35,36,37].

The synthesized particles were shown to be effective theranostic agents, realizing effective magnetically assisted targeted delivery, tumor bioimaging, and treatment under external light irradiation. The in vitro and in vivo functionality of these nanoparticles were thoroughly tested, and 100% inhibition of allograft solid tumor growth was shown, thus confirming the great potential of the developed nanoformulation for bioimaging and PDT.

## 2. Materials and Methods

### 2.1. Magnetite Synthesis

The magnetite cores incorporated into PLGA were synthesized as described by us previously with some modifications [9,38,39]. A total of 0.86 g FeCl_2_·4H_2_O and 2.36 g FeCl_3_·6H_2_O were dissolved in 40 mL of Milli-Q water, then 3 mL of 25% NH_4_OH was added, and the resulting mixture was rapidly stirred. The mixture was then incubated in a water bath at 80 °C for 2 h with subsequent cooling to room temperature. The resulting magnetic fraction was washed using the magnetic separation and then 10 mL of 0.5 M HNO_3_ was added, with the mixture incubated for 5 min and magnetically separated through the magnet application for 5 min. The supernatant was then removed, and 10 mL of H_2_O was added to the magnetic fraction. Then, this fraction was sonicated and magnetically separated again. The procedure was repeated thrice, and 3 magnetic fractions were collected.

Next, 600 µL of the third magnetite fraction was mixed with 2 mL of oleic acid and 5 mL of chloroform and sonicated for 3 min. Next, 15 mL of chloroform was added, and the mixture was sonicated again for 2 min. After that, 300 µL of 2M NaOH was added, and the mixture was sonicated again for 1 min. The oleic-acid-stabilized magnetite was then centrifuged for 1 h at 20,000× *g* at 20 °C, and the resulting mixture was resuspended in 3 mL of dichloromethane.

### 2.2. mPLGA/IR775 Nanoparticle Synthesis

Hybrid magnetic polymer nanoparticles were synthesized by the “oil-in-water” microemulsion method followed by solvent evaporation based on the method previously described by us with modifications [40,41,42]. A total of 12 mg PLGA (RG 858 S, lactide/glycolide 85:15, MW 190–240 kDa, Sigma, Darmstadt, Germany), 100 µL of oleate-stabilized magnetite, and 250 µg IR775 (Sigma, Darmstadt, Germany) in 300 µL of dichloromethane were added to 3 mL of 3% aqueous PVA (Mowiol 4-88, Sigma, Darmstadt, Germany) supplemented with chitosan oligosaccharide lactate with a final concentration of 1 g/L (5 kDa, Sigma, Darmstadt, Germany). The mixture was treated with ultrasound for 1 min. After solvent evaporation, the particles were washed thrice with centrifugation for 5 min at 5000× *g* and finally resuspended in 10 mM HEPES (pH 7.0).

### 2.3. Electron Microscopy Analysis

The morphology of nanoparticles was studied with a MAIA3 Tescan (Tescan, Brno-Kohoutovice, Czech Republic) scanning electron microscope. The sample of nanoparticles at 10 µg/mL was applied on a silicon wafer on carbon film and air-dried, followed by SEM examination at an accelerating voltage of 7 kV. Magnetite was analyzed with transmitting electron microscopy as well at an accelerating voltage of 70 kV (JEOL JEM2100Plus transmitting electron microscope). The scanning electron microscopy images were processed in ImageJ as follows: the sizes of 250 nanoparticles were measured, and then the average particle size was calculated.

### 2.4. Spectroscopy

The absorbance spectra of nanoparticles were measured with the CLARIOstar microplate reader (BMG Labtech, Ortenberg, Germany) within the 300–1000 nm range for particles at 0.3 g/L in H_2_O with subsequent subtraction of the absorbance spectrum of pure H_2_O.

Fluorescent excitation and emission spectra were registered with an Infinite M1000 Pro microplate reader (Tecan, Grödig, Austria). The excitation spectrum was recorded at λem = 800 nm within the 400–785 nm range, and the emission was recorded at λex = 700 nm within the 715–850 nm range. mPLGA/IR775 nanoparticles at 0.3 g/L in water were used for analysis. The obtained spectra were normalized. The dye loading efficiency was calculated by measuring the fluorescence intensity of mPLGA/IR775 nanoparticles at 0.3 g/L in a 50%/50% DMSO/H_2_O mixture with λex = 730 nm and λem = 800 nm. A calibration curve for free IR775 in the 50%/50% DMSO/H_2_O mixture was used to determine dye loading content.

The generation of reactive oxygen species was measured as follows: nanoparticles at 1 g/L in H_2_O were mixed with CM-H_2_DCFDA to obtain a final concentration of 5 µM, irradiated with 808 nm laser for 5 min, and centrifuged for 5 min at 10,000× *g*; following this, the fluorescence of 100 µL of supernatant was measured with λex = 500 nm and λem = 525 nm using a CLARIOstar microplate reader (BMG Labtech, Ortenberg, Germany), and the autofluorescence of blank wells was subtracted.

### 2.5. DLS Measurements

The hydrodynamic sizes and ζ-potentials of nanoparticles were analyzed using ZetaSizer Nano ZS (Malvern Instruments Ltd., Worcestershire, UK) in 10 mM HEPES (pH 7.0).

### 2.6. Chemical Conjugation

mPLGA/IR775 nanoparticles were conjugated to proteins using carbodiimide chemistry with EDC/sulfo-NHS as crosslinking agents. A total of 10 mg of mPLGA/IR775 were incubated with 5 mg EDC and 0.5 mg sulfo-NHS in 200 µL of 0.1 M MES buffer for 20 min at room temperature. Next, nanoparticles were purified from the excess of chemicals through centrifugation for 5 min at 5000× *g*. Following this, 200 µL of protein at 1 g/L in 0.1 M HEPES (pH 6.0) was added to the nanoparticles, and the suspension was sonicated for 10 s and incubated for at least 4 h at room temperature. Then, nanoparticles were purified from non-bound protein with triple centrifugation and resuspended in 10 mM HEPES (pH 7.0).

### 2.7. Cell Culture

EMT6/P, EA.hy926, and NIH/3T3 cells (Shemyakin-Ovchinnikov Institute RAS, Molecular Immunology Laboratory collection) were cultured in DMEM medium (HyClone, Logan, UT, USA) supplemented with 10% FBS (HyClone, Logan, UT, USA), penicillin/streptomycin (PanEko, Moscow, Russia), and 2 mM L-glutamine (PanEko, Moscow, Russia) at 37 °C and 5% CO_2_.

### 2.8. Flow Cytometry

ROS generation was assessed with the flow cytometry test. A total of 0.2 × 10^6^ EMT6/P cells in 200 µL PBS with 1% BSA were incubated with 0.2 g/L of mPLGA/IR775 nanoparticles and ROS sensor according to the manufacturer recommendations (Total Reactive Oxygen Species (ROS) Assay Kit 520 nm, Invitrogen, Thermo Fisher Scientific Inc., Waltham, MA, USA) for 30 min at 37 °C. Next, cells were washed from non-bound nanoparticles by centrifugation for 5 min at 100× *g* and irradiated with an 808 nm laser (600 mW) for 2 min. Thirty minutes later, the fluorescence of cells was analyzed with the Novocyte 3000 VYB flow cytometer (ACEA Biosciences, San Diego, CA, USA) in the BL1 channel (excitation laser 488 nm, emission filter 530/30 nm).

Particle binding efficiency was studied as follows. 0.2 × 10^6^ cells in 200 µL PBS with 1% BSA were incubated with nanoparticle conjugates, then washed from non-bound nanoparticles by centrifugation for 5 min at 100× *g*, and the fluorescence of cells was analyzed with the Accuri C6 (BD) flow cytometer in the FL4 channel (λex = 640 nm and λem = 675/25 nm).

### 2.9. Fluorescent Microscopy

EMT6/P cells were seeded on 96-well plates at 20 × 10^3^ cells per well in 150 µL of full culture media and cultured overnight. Next, nanoparticles were added to wells to obtain a final concentration of 50 µg/mL, and plates were incubated for 30 min at 37 °C and 5% CO_2_. Next, wells were washed thrice with full culture media and samples were analyzed using an epifluorescent Zeiss microscope at the following conditions: λex = 595–645 nm, λem = 670–725 nm.

### 2.10. Cell Toxicity Study

The cytotoxicity of nanoparticles was investigated using a resazurin-based cytotoxicity test. A total of 2 × 10^6^ EMT6/P cells in 1 mL of full culture media were incubated with nanoparticles at different concentrations for 30 min at 37 °C with 5% CO_2_. Then, the cells were washed from non-bound nanoparticles by centrifugation for 3 min at 100× *g*. Next, cells were irradiated with an 808 nm laser (600 mW) for various time intervals. The cells were then diluted with full culture medium, and 5 × 10^3^ cells in 150 µL of media were seeded into the wells of 96-well plates and cultured for 72 h at 37 °C with 5% CO_2_. Then, the medium was removed from the wells, 100 µL of resazurin solution (13 mg/L in PBS) was added, and samples were incubated for 3 h at 37 °C and 5% CO_2_. The fluorescence of wells was then measured using the CLARIOstar microplate reader (BMG Labtech, Ortenberg, Germany) at wavelengths of λex = 570 nm and λem = 600 nm. Data are presented as percentages from non-treated and non-irradiated cells.

### 2.11. Tumor Bearing Mice

Female BALB/c mice of 22–25 g weight were purchased from the Puschino Animal Facility (Shemyakin-Ovchinnikov Institute of Bioorganic Chemistry Russian Academy of Sciences, Pushchino branch of the Institute, Pushchino, Russia) and maintained at the Vivarium of the Shemyakin-Ovchinnikov Institute of Bioorganic Chemistry Russian Academy of Sciences (Moscow, Russia). All procedures were approved by the Institutional Animal Care and Use Committee (IACUC) of the Shemyakin-Ovchinnikov Institute of Bioorganic Chemistry Russian Academy of Sciences (Moscow, Russia) according to the IACUC protocol # 299 (1 January 2020–31 December 2022).

The animals were anesthetized with a mixture of Zoletil (Virbac, Carros, France) and Rometar (Bioveta, Ivanovice na Hané, Czech Republic) at a dose of 25/5 mg/kg.

To create tumor allografts, mice were injected with 4 × 10^6^ EMT6/P cells in 100 µL of culture media into the right flank. The tumor volume was measured with a caliper and calculated using the following formula V = a^2^·b/2, where a is the tumor width and b is the tumor length.

### 2.12. MPQ-Measurements

BALB/c mice with EMT6/P tumor were i.v. injected with 1 mg of nanoparticles through the retroorbital sinus injection with or without magnet application to the tumor site. Then, 24 h later, animals were euthanized with cervical dislocation, and organs were extracted, weighed, and fixed in the 4% formalin. The organ magnetic signal was measured using our original MPQ device [32,38].

### 2.13. Bioimaging Study

For bioimaging experiments, mice were anesthetized with tiletamine-HCl/zolazepam-HCl/xylazine-HCl and then placed into a chamber of LumoTrace FLUO bioimaging tomograph (Abisense, Sochi, Russia). The images were acquired with fluorescence excitation = 730 nm and 780LP nm emission filter and exposure of 1000 ms.

### 2.14. In Vivo PDT

For photodynamic therapy in vivo, mice with tumors of about 190 ± 67 mm^3^ were selected and randomly divided into 3 groups (n = 6) and treated as follows: the first group served as the control non-treated cohort, and the second and third groups received the i.v. injections of 1 mg of nanoparticles with the magnet application to the tumor site. Then, 24 h later, mice from the third group only were irradiated with an 808 nm laser (600 mW) for 30 min. The dynamics of tumor growth were then measured every 2 days with a caliper.

## 3. Results

### 3.1. Design of the Experiment

The efficient PDT agent based on a polymer matrix was designed as follows (Figure 1): PLGA-based nanoparticles loaded with magnetite and a photodynamic sensitizer, IR775 dye, were synthesized (mPLGA/IR775). Magnetite in the composition of these nanoparticles makes it possible to quantitatively trace particle accumulation in the organism and realize magnetic-assisted drug delivery, while IR775 dye allows for the visualization of nanoparticle accumulation non-invasively in vivo, and, under external light irradiation, producing ROS, thus performing PDT.

Next, these nanoparticles were equipped with a spectrum of plant lectins using standard chemical conjugation. BSA-conjugated PLGA nanoparticles and unmodified PLGA nanoparticles were used as control samples. The efficiency of binding of these conjugates to cells was then screened by flow cytometry. Three types of cell lines were studied: cancer cells (mouse mammary breast cancer cells, EMT6/P), immortalized fibroblasts as a model of normal cells (NIH/3T3 cells), and immortalized endothelial cells (EA.hy926). Those particles were selected that have the maximum binding to cancer cells and endothelial cells while having a minimal effect on non-transformed fibroblast cells. Flow cytometry tests showed that such a leader is nanoparticles equipped with concanavalin A (mPLGA/IR775-ConA).

These nanoparticles were i.v. injected into mice with allograft tumors without and with magnetic delivery by applying a magnet to the site of the tumor. In vivo imaging tests showed a significant accumulation of nanoparticles in the tumor site, and exposure to the 808 nm laser led to complete remission in all mice from the light-treated group, thus realizing the concept of theranostics, namely, the diagnostics and the therapy using the same nanoformulation.

### 3.2. Synthesis and Characterization of Magnetite-Loaded PLGA Nanoparticles

PLGA nanoparticles loaded with magnetite and photodynamic sensitizer IR775 dye were synthesized by the “oil–water” microemulsion method as schematically shown in Figure 1a. Prior to magnetic polymer nanoparticle synthesis, magnetic cores were synthesized and stabilized for the effective incorporation into the PLGA matrix.

Magnetite was synthesized according to the protocol in the Materials and Methods section using the standard co-precipitation technique with some modifications. The size of these magnetic cores determined by scanning electron microscopy was found to be 14.7 ± 5.5 nm, having a form close to spherical (Figure 2a,c).

Next, particles were coated with sodium oleate/oleic acid. It was found that the particles coated with oleic acid were more stable than those coated with sodium oleate. Sodium hydroxide solution was added to the obtained magnetite to stabilize it, which resulted in the colloidal stability of the magnetite for at least one month.

The resultant oleic-acid-stabilized magnetic nanoparticles were used for PLGA microemulsion synthesis. As synthesized mPLGA/IR775 nanoparticles were analyzed with scanning electron microscopy, which confirmed the incorporation of magnetite. The analysis of nanoparticle sizes based on SEM image processing showed that particles were 281.2 ± 83.9 nm with spherical form (Figure 2b,d).

The absorbance spectra of mPLGA/IR775 as well as non-magnetic PLGA/IR775 nanoparticles are presented in Figure 2e: the pronounced peak near the 800 nm region corresponded to the incorporated IR775 dye in the nanoparticle structure, while the decrease in the spectrum intensity for mPLGA/IR775 nanoparticles in the UV region most likely occurred due to the increased light scattering by the magnetic cores within the nanoparticle.

The effective incorporation of IR775 to mPLGA/IR775 nanoparticles was confirmed by measuring the excitation and emission fluorescence spectra of as-synthesized nanoparticles (Figure 2f). Data presented in Figure 2f confirm that the fluorescence of nanoparticles corresponded to the expected fluorescence of nanoparticles according to IR775 dye loading.

The measurement of IRR75 loading efficiency showed that the dye loading was equal to 16.8 ± 0.8 µg of IR775 per 1 mg of mPLGA/IR775, thus demonstrating 80.6% loading efficiency during the synthesis (250 µg of IR775 were used in synthesis per 12 mg of PLGA).

The effective reactive oxygen species (ROS) production of PLGA/IR775 and mPLGA/IR775 under light irradiation was evaluated using a CM-H_2_DCFDA sensor pre-mixed with nanoparticles. The mixture was light-irradiated, particles were centrifuged, and the fluorescence of the supernatant was measured (Figure 3a). The data presented in Figure 3a confirm that particles produced ROS under light irradiation. Interestingly, the ROS production was 2.5 higher for nanoparticles loaded with magnetite and IR775 in comparison with particles loaded with IR775 only under light irradiation (Figure 3a).

Next, ROS generation was studied with flow cytometry tests. EMT6/P cells were incubated with mPLGA/IR775 nanoparticles and a total ROS sensor. Cells were washed from non-bound nanoparticles, then irradiated with an 808 nm laser and analyzed with flow cytometry in the fluorescent channel corresponding to ROS sensor fluorescence. The data presented in Figure 3b confirm that mPLGA/IR775 are effective PDT sensitizers since these particles generate ROS under light irradiation.

### 3.3. Chemical Conjugation of mPLGA/IR775 to Lectins

As-synthesized mPLGA/IR775 were conjugated to different plant lectins using standard carbodiimide chemistry with EDC/sulfo-NHS as cross-linking agents. Ulex Europaeus Agglutinin (UEA), soybean agglutinin (SBA), peanut agglutinin (PNA), and concanavalin A lectin from Canavalia ensiformis (Con A) were used for chemical conjugation. Particles conjugated with BSA as well as pristine non-conjugated nanoparticles served as control nanoparticles for the in vitro tests.

The stability of mPLGA/IR775–lectin conjugates was studied both visually and with the dynamic light scattering method.

Cumulant analysis showed the size of nanoparticles and their conjugated to be 314.3 ± 114.6 nm for mPLGA/IR775, 258.7 ± 80.67 nm for mPLGA/IR775-BSA, 442.3 ± 206.4 nm for mPLGA/IR775-UEA, 274.1 ± 94.63 nm for mPLGA/IR775-SBA, 276.0 ± 111.1 nm for mPLGA/IR775-PNA, and 279.4 ± 107 nm for mPLGA/IR775-ConA (Figure 3c), thus confirming the difference between conjugated and non-conjugated nanoparticles and proving their colloidal stability.

Electrophoretic light scattering measurements showed nanoparticle ζ-potentials to be equal to −20.2 ± 10.4 mV for mPLGA/IR775, −21.3 ± 4.84 mV for mPLGA/IR775-BSA, −19.9 ± 6.42 mV for mPLGA/IR775-UEA, −20.3 ± 6.91 mV for mPLGA/IR775-SBA, −19.2 ± 6.05 mV for mPLGA/IR775-PNA, and −17.7 ± 5.99 mV for mPLGA/IR775-ConA. Since the isoelectric points of conjugated lectins are in the slightly acidic region, pI 4.5−5.1 for UEA, pI 5.8–6.0 for SBA, pI 5.5–6.5 for PNA, and pI 4.5–5.5 for ConA, all of the studied lectins were charged neutrally or slightly negatively in buffers with pH 7.0. A possible problem would be that when a significant number of proteins are conjugated to the nanoparticle surface, the particles can aggregate due to a strong increase in surface charge. However, measurements showed only a slight increase in ζ-potentials due to the optimal concentration of proteins during conjugation, and the particles retained colloidal stability.

### 3.4. mPLGA/IR775–Lectin Interaction with Cells: Flow Cytometry Screening and In Vitro Cytotoxicity Tests

The obtained spectrum of synthesized nanoparticles modified with lectins was studied for interaction with different cell lines by flow cytometry. Thus, the binding of PLGA conjugates to the following cell lines was studied: (1) EMT6/P mouse breast cancer cells, (2) vascular endothelial cells EA.hy926, (3) non-cancerous cells—immortalized fibroblasts NIH/3T3 cells. The aim of this test was to find such conjugates that would more efficiently bind to cancer breast cells (EMT6/P) and endothelial cells (EA.hy926) to block tumor angiogenesis processes and at the same time have minimal binding to non-transformed cells—in this case, fibroblasts—for minimizing the negative effect on healthy tissues in the organism.

The described cells were incubated with the nanoparticle conjugates, washed from non-bound nanoparticles, and analyzed by flow cytometry in the fluorescence channel corresponding to the fluorescence of nanoparticles (Figure 4a). The binding efficiency was quantified using the *stain index* calculated as *stain index = (MFI_pos_ – MFI_auto_)/2SD_auto_*, where *MFI_pos_* and *MFI_auto_* are medians of the fluorescence intensity of labeled and non-labeled cell populations, respectively, and *SD_auto_* is the standard deviation of *MFI_auto_*. The obtained calculations are shown in Figure 4b. The data presented indicate that it was the conjugates of nanoparticles with ConA that most effectively bound to mouse breast cancer cells EMT6/P and endothelial cells EA.hy926 while minimally affecting fibroblasts NIH/3T3. Thus, we suggested that mPLGA/IR775-ConA are leader nanoparticles for breast cancer PDT.

Next, we studied the binding of these nanoparticles to cells by fluorescence microscopy. The data presented in Figure 4c indicate a more efficient penetration of mPLGA/IR775-ConA nanoparticles into cells compared to control mPLGA/IR775-BSA nanoparticles. Since breast cancer cells overexpressed more mannose N-glycan units on the membrane proteins than that of normal cells, the binding of mPLGA/IR775-ConA occurred due to the interaction of ConA with mannose residues on several proteins types in cancer cells. N-glycosylation, the sequential addition of complex sugars to adhesion proteins, ion channels, and receptors, is one of the most frequent protein glycosylation processes, leading to the fact that ConA-based nanoparticles interact both with internalizing and non-internalizing proteins on the cell membrane, resulting in the nanoparticle accumulation in different cell compartments, resting on the cell membrane, localizing in endosomes and in the cytoplasm [43,44,45]. Maybe this feature is one of the most interesting aspects of lectin-modified nanoparticles’ interaction with cancer cells gathering the attention of researchers, since this fact allows for the affecting several types of cell compartments simultaneously, which is very important on the road to the development of multifunctional cancer-fighting strategies, especially those based on PDT.

The effective interaction of nanoparticles with cells made it possible to achieve effective light-induced cytotoxicity. Thus, EMT6/P cells were incubated with nanoparticles, washed from non-bound ones, and irradiated with an 808 nm laser. Seventy-two hours later, a resazurin-based test was performed, which showed that, in a wide range of concentrations, nanoparticles are cytotoxic only when exposed to external radiation, and the cytotoxicity depends on the irradiation time (Figure 4d).

### 3.5. Diagnostic Properties of mPLGA/IR-775-ConA Nanoparticles: In Vivo and Ex Vivo Biodistribution Study

As-synthesized hybrid polymer magnetic nanoparticles of mPLGA/IR775-ConA were then i.v. injected into tumor-bearing mice in order to study their diagnostic and therapeutic capabilities. BALB/c mice with EMT6/P allograft tumors in the right flanks were used for this study. The magnetically guided targeted delivery of nanoparticles was realized, and 24 h after the nanoparticle injection, the accumulation of nanoparticles was studied both in vivo and ex vivo.

In vivo bioimaging was performed using the fluorescent properties of IR775 dye inside the nanoparticles using the LumoTrace Fluo (Abisense, Sochi, Russia) bioimaging device. Two targeting modes were compared, namely, magnetically guided delivery with the magnet applied near the tumor site and delivery without the applied magnet. Figure 5a presents fluorescent images of mice that were injected with nanoparticles without and with the use of an external magnetic field. These images indicate an increase in delivery efficiency when exposed to a magnetic field. In vivo images were accompanied by ex vivo photographs of the extracted organs, and the intensity of fluorescence of tumors in mice from the group with a magnet was to a certain degree higher.

Since fluorescent images do not provide information on the quantitative biodistribution of nanoparticles in mice organs and are made available only a qualitative picture of particle biodistribution, we studied the accumulation of nanoparticles in organs using our original MPQ method (magnetic particle quantification) [38] to assess the efficacy of magnetic delivery. MPQ is an efficient method for the quantitative measurement of the accumulation of magnetic nanoparticles in the organism. The method has a sensitivity limit of 0.33 ng nanoparticles and is devoid of background signals from liquids and tissues; each measurement takes a few seconds and does not require complex sample preparation or data processing. This is possible due to the way the device works: a magnetic coil generates an alternating electromagnetic field with two components, thereby affecting magnetic particles. Linear magnetic materials (dia- and paramagnetics) respond only at the frequencies of the applied field, whereas non-linear magnetic materials (e.g., superparamagnetic materials) respond at the combinatorial frequencies of the field [32,38,46].

Extracted organs of mice from two groups: with and without magnetic delivery (n = 10 in each group) were measured with the MPQ device in order to obtain particle distribution in organs. The data presented in Figure 5b demonstrate the difference in nanoparticle accumulation in tumors for these groups: 4.5 ± 1.5% vs. 7.3 ± 1.3% of the injected dose for groups without and with the applied magnetic field, respectively. Significant differences between the control group and experimental group determined with two-sided unequal variances t-tests (Welch’s test) for two-group comparisons showed the p-value to be equal to 0.0004, thus proving the difference in nanoparticle accumulation in tumors.

Here, we showed that the application of a magnetic field increased the delivery of mPLGA/IR775-ConA nanoparticles into the tumor by 1.6 times. We should emphasize the problem that always arises on the way to the development and verification of human anti-cancer drugs using laboratory mice. Since mice are small rodents and their tumors are correspondingly small, it is quite difficult to predict how targeted nanoparticles will behave in terms of quantitative accumulation in the large human tumor, especially considering the fact that there are no successful cases of introducing targeted nanoparticles into the clinic and there is no way to obtain relevant data from the literature for at least one of the existing targeted nanoparticles. We can only try to model the processes in rodents that maximally reflect the processes of carcinogenesis in humans. For this purpose, the allograft tumor models were used in this study, namely, mouse cancer cells were injected in mice, while the tumors formed for a rather long time period (for almost 2 weeks) and were allowed to grow to a fairly large size, 190 ± 67 mm^3^ (thus, corresponding to approximately 1% from total body weight), developing a normal vasculature, thereby minimizing the EPR effect of fast-growing tumors such as in B16 or CT26 models.

However, it is rather presumptuous to suggest that such efficiency and such a difference in the accumulation of nanoparticles with and without a magnetic delivery can be reproduced in human tumors. To shed light on this problem, it is necessary to perform a series of sequential experiments using larger mice, e.g., ICR (CD-1) instead of Balb/c, large rats, and other larger animals, and this is included in our plans for further research.

### 3.6. Therapeutic Properties of Lectin-Modified Nano-PLGA

The therapeutic properties of lectin-conjugated magnetic polymer nanoparticles were then studied. Mice with allograft EMT6/P tumors were randomly divided into three groups when the tumor sizes reached 190 ± 67 mm^3^ (n = 6). Mice were treated as follows: (i) the first group served as the control non-treated cohort, (ii) the second group was treated with mPLGA/IR775-ConA nanoparticles, and (iii) the third group was treated with mPLGA/IR775-ConA nanoparticles followed by light irradiation. Mice received single i.v. injections of 1 mg of mPLGA/IR775-ConA nanoparticles with a magnet applied to the tumor site. Twenty-four hours later, the tumor sites of mice from the third group were irradiated with an 808 nm laser for 30 min (Figure 6b). The tumor growth dynamics that were monitored by caliper measurements every two days are presented in Figure 6d. Data presented in Figure 6a–d confirm the efficacy of mPLGA/IR775-ConA as tumor-targeting agents for light-PDT: tumor growth inhibition was found to be 100% in the light-treated group with complete tumor elimination.

## 4. Discussion

The main goal of this work was the development of the most effective nanoagent for large solid tumor PDT. We synthesized polymer nanoparticles loaded with magnetite and the PDT agent, IR775 dye (mPLGA/IR775). The magnetite incorporation into PLGA particle structure allows for quantitative tracking of their accumulation in different organs and performing magnetic-assisted delivery, while IR775 makes fluorescent in vivo bioimaging possible, as well as light-induced PDT. To equip PLGA nanoparticles with targeting modality, the particles were conjugated with lectins of different origins, and the flow cytometry screening revealed that the most effective candidate for breast cancer cell labeling is ConA, a lectin from *Canavalia ensiformis*.

PDT is one of the most promising cancer treatment strategies, being based on the absorption of light by a photosensitizer and the conversion of oxygen into reactive oxygen species (ROS). Previously, various scientific groups have shown that PLGA nanoparticles are excellent nanoagents for PDT. Thus, different PLGA-based formulations loaded with photosensitizers were developed: Rose Bengal [42], indocyanine green [47], zinc phthalocyanine [48,49], and others, and their applications in vitro and in vivo were demonstrated for selective cancer cell destruction under the external light source. For example, Fadel et al. showed the efficacy of PLGA loaded with zinc(II) phthalocyanine (ZnPc) in vivo using Ehrlich ascite carcinoma cells under 880 nm light irradiation [48]. Zhang et al. demonstrated both the bioimaging capabilities of IR775-loaded PLGA using BALB/c mice with CT26 tumors and showed that the therapeutic modalities of particles led to the tumor growth inhibition of 87.28% [50]. The heptamethine cyanine derivative, IR775 dye (2-[2-[2-chloro-3-[2-(1,3-dihydro-1,3,3-trimethyl-2H-indol-2-ylidene)-ethylidene]-1-cyclohexen-1-yl]-ethenyl]-1,3,3-trimethyl-3H-indolium chloride), was shown to be one of the most effective PDT sensitizers, especially within the composition of different nanoparticles due to its hydrophobic nature [35,36,37].

To achieve targeted delivery, we chemically conjugated PLGA nanoparticles with concanavalin A. ConA is a lectin purified from jack bean (*Canavalia ensiformis*) that specifically binds D-mannose/D-glucose residues on the cell surface [51,52,53]. Several studies confirmed the ability of ConA-coated nanoparticles of various natures to preferentially bind to cancer cells. Thus, Chen et al. demonstrated that the binding capacity of ConA-conjugated silica–carbon hollow spheres was higher for liver cancer cells than for normal cells in vitro [54]. Martínez-Carmona et al. reported that ConA-conjugated mesoporous silica nanoparticles loaded with doxorubicin specifically bind and kill human osteosarcoma cells in contrast to healthy preosteoblast cells, MC3T3-E1 [55]. Chowdhury et al. showed ConA-conjugated quantum dots loaded with doxorubicin possess higher specific cytotoxicity against HeLa cancer cells compared to normal cells in vitro [53]. Khopade et al. reported ConA-coated multiple emulsion bearing 6-mercaptopurine [56] outperformed uncoated emulsion (ME) and free drug with IC50 0.7, 2.5, and 2.8 µM on murine leukemia cell line L-1210 in vitro. The same study demonstrated that the median survival time of mice treated with ConA ME was superior to that of ME and free drug [56]. In addition, it was shown that concanavalin A can modulate signaling pathways, inducing apoptosis in melanoma A375 cells and autophagy in glioblastoma and hepatoma cells, as well as inhibit proliferation of melanoma B16 cells and fibroblast 3T3 cells [30,57,58,59]. Hence, concanavalin A is a promising molecule that can not only recognize cancer cells but also lead to the suppression of their vital activity.

It should be also highlighted that ConA, like most lectins, is a homotetramer with one subunit of 26.5 kDa, and thus problems with steric hindrances that constantly arise during the conjugation of nanoparticles with antibodies or peptides are much less pronounced. Even if one of the homotetramer subunits is attached to the surface of the particle, the other three will still be active. In this way, most of the standard problems of chemical conjugation are removed, and the optimization of ligation protocols requires much less effort.

The combination of photodynamic properties, magnetite incorporation, and lectin modification within the single PLGA matrix allows for the obtaining of trifunctional nanoparticles for magnetically assisted targeted drug delivery. Such hybrid mPLGA/IR775-ConA nanoparticles were shown to (i) efficiently target the allograft tumors, (ii) perform magnetically assisted delivery (the efficacy of the delivery was 1.6 times greater under the application of the magnetic field), and (iii) induce tumor elimination under external light irradiation with tumor growth inhibition = 100% in all mice from the treated group.

## 5. Conclusions

This study is a step forward in the development of new-generation nanomedications for cancer PDT. Biocompatible PLGA nanoparticles are already widely used in clinical applications as well as some methods of PDT. We believe that the combination of lectin-assisted magnetic targeted delivery and all the advantages of PLGA carriers and PDT efficacy would result in a new smart nanoagent for oncotheranostics.

## Figures and Tables

**Figure 1 pharmaceutics-15-00092-f001:**
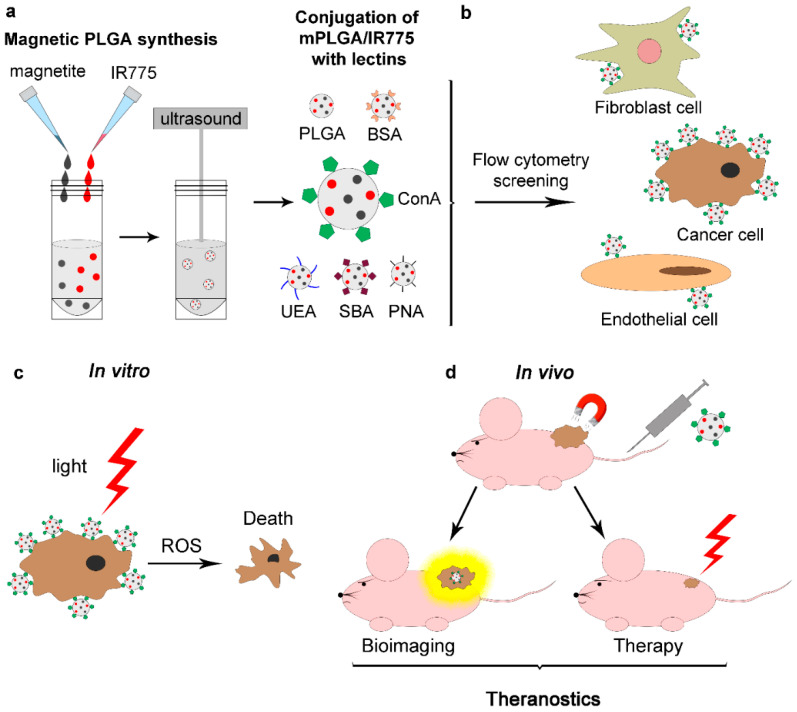
Lectin-conjugated magnetic PLGA for photodynamic therapy: scheme of the experiment. (**a**) Magnetite and photosensitizer IR775-loaded PLGA nanoparticles were synthesized via the “oil-water” microemulsion method. (**b**) As-synthesized mPLGA/IR775 nanoparticles were conjugated with different lectins and screened with flow cytometry for the most effective binding to cancer cells. (**c**) In vitro cell toxicity studies: cancer cells labeled with mPLGA/IR775-ConA were irradiated with an external IR light leading to cancer cell death. (**d**) In vivo diagnostics and therapy study: magnet-assisted delivery of mPLGA/IR775-ConA allowed for effective visualization and elimination of the tumor, thus realizing the theranostics concept.

**Figure 2 pharmaceutics-15-00092-f002:**
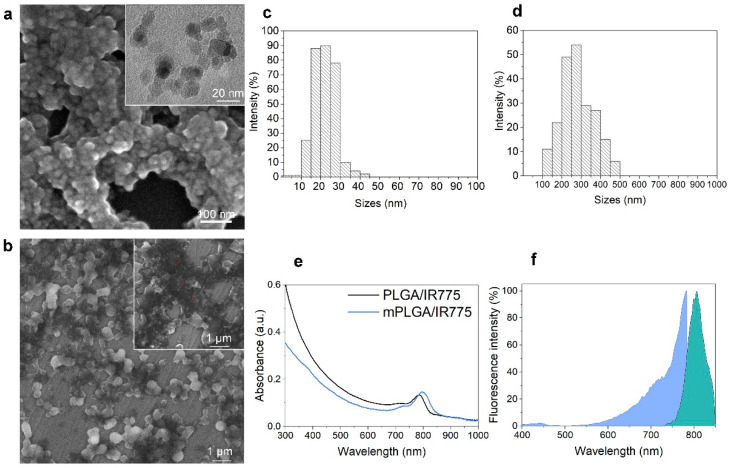
Synthesis and characterization of mPLGA/IR775 nanoparticles. (**a**) Scanning electron microscopy of magnetite synthesized with the coprecipitation technique. The inset is transmitting electron microscopy of magnetite. (**b**) Scanning electron microscopy of PLGA nanoparticle loaded with magnetite and IR775 photosensitizer. The SEM inset demonstrates the incorporation of magnetite: red arrows show white dots corresponding to magnetite cores. (**c**) Physical size distribution of magnetite obtained via SEM image processing. (**d**) Physical size distribution of PLGA nanoparticles loaded with magnetite and IR775 (mPLGA/IR775). (**e**) UV–VIS absorbance spectrum of mPLGA/IR775 and PLGA/IR775 nanoparticles. (**f**) Excitation and emission fluorescence spectra of mPLGA/IR775 nanoparticles: blue—fluorescence excitation (measured with λem = 800 nm), fluorescence emission (measured with λex = 700 nm).

**Figure 3 pharmaceutics-15-00092-f003:**
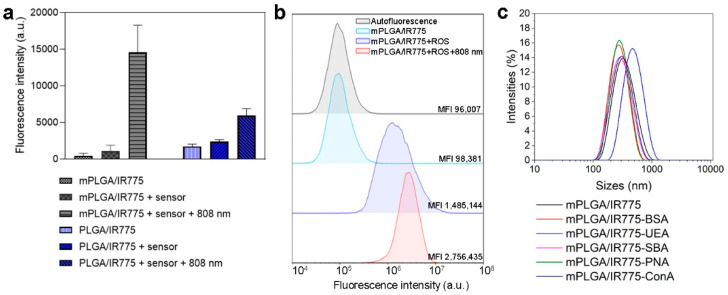
Characterization of mPLGA/IR775 nanoparticles as nanoagents for PDT. (**a**) ROS generation study of nanoparticles under the irradiation with an 808 nm laser: magnetic mPLGA/IR775 and non-magnetic PLGA/IR775 nanoparticles were incubated with ROS sensor and irradiated with 808 nm light and centrifuged, and the fluorescence of the supernatant was measured (λex = 500 nm, λem = 525 nm). (**b**) Flow cytometry study of ROS generation. Cells were incubated with mPLGA/IR775 nanoparticles and ROS sensor and irradiated with 808 nm laser for 10 min. Histograms represent cell populations in the fluorescent channel corresponding to the ROS sensor fluorescence (λex = 488 nm, λem = 525/20 nm). (**c**) mPLGA/IR775 were conjugated to different lectins as well as to BSA and characterized with the dynamic light scattering method, confirming the colloidal stability of nanoparticles after chemical conjugation.

**Figure 4 pharmaceutics-15-00092-f004:**
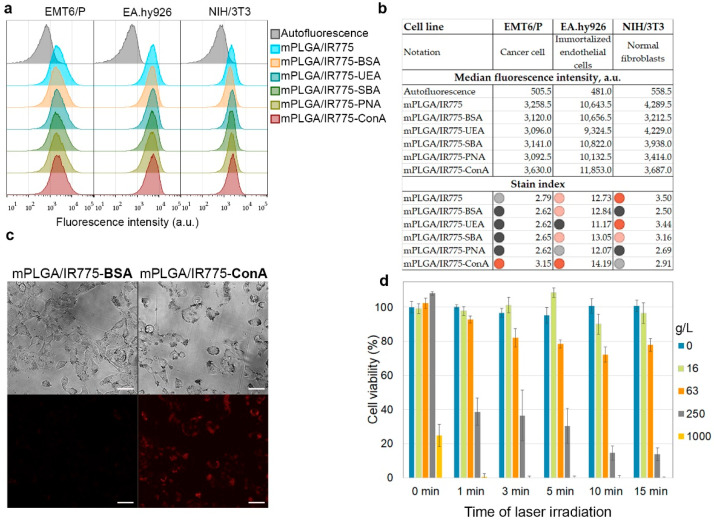
In vitro study of mPLGA/IR775 interaction with cells. (**a**) Flow cytometry assay on evaluation of the binding efficiency of lectin-modified nano-PLGA. EMT6/P cells were labeled with lectin-conjugated particles and analyzed with flow cytometry in the fluorescence channel corresponding to IR775 fluorescence. (**b**) Flow cytometry histograms are accompanied by the median fluorescence intensity data and stain index calculation for three cell lines. (**c**) Fluorescent microscopy of EMT6/P cell labeled with mPLGA/IR775-BSA and mPLGA/IR775-ConA nanoparticles. Scale bars, 50 µm. (**d**) Resazurin test on the evaluation of mPLGA/IR775-ConA nanoparticle cell toxicity under 808 nm laser light irradiation.

**Figure 5 pharmaceutics-15-00092-f005:**
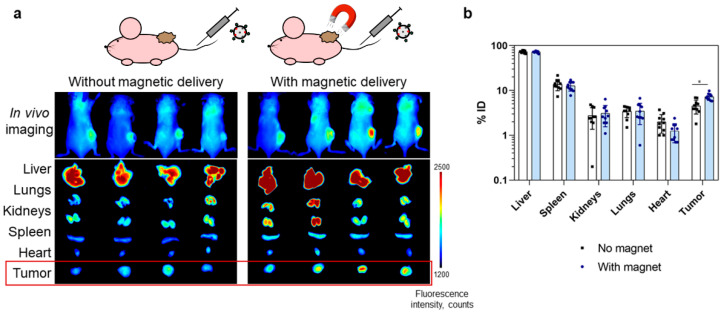
Diagnostic properties of mPLGA/IR-775-ConA nanoparticles: in vivo imaging and ex vivo evaluation of particle accumulation in tumors with and without magnet-assisted delivery. (**a**) Fluorescent images of mice 24 h after the i.v. injection of mPLGA/IR775-ConA with and without magnetic targeting. Data are accompanied by the ex vivo fluorescent images of mice organs (λex = 730 nm, λem = 780 nm). (**b**) Percent of injected dose accumulated in the main organs according to MPQ measurements. Data are presented on a logarithmic scale (n = 10 for each group). * *p* < 0.001.

**Figure 6 pharmaceutics-15-00092-f006:**
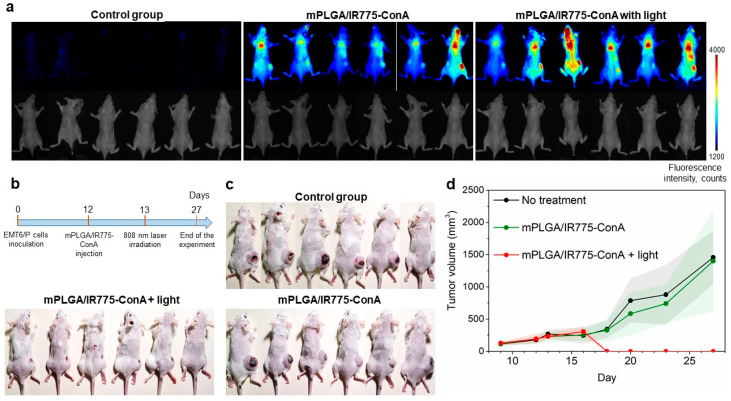
Therapeutic properties of lectin-modified nano-PLGA: mPLGA/IR775-ConA sensitizer-assisted photodynamic therapy of EMT6/P tumors. Mice were i.v. injected with 1 mg of mPLGA/IR775-ConA nanoparticles and 24 h later were irradiated with an 808 nm light (600 mW). (**a**) Evaluation of tumor targeting ability of mPLGA/IR775-ConA nanoparticles: fluorescent images of mice 24 h after the injection of nanoparticles (control group, group that received the nanoparticle injection only, and group that afterward was treated with light irradiation). (**b**) Scheme of the treatment. (**c**) Smartphone camera images at the end of the experiment (day 27): images of mice from all treated groups (n = 6). (**d**) Tumor growth dynamic of mice treated with mPLGA/IR775-ConA nanoparticles.

## Data Availability

Samples of nanoparticles are available from the authors.

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
