# Peer review of "Lectin-Modified Magnetic Nano-PLGA for Photodynamic Therapy In Vivo"

_pharmaceutics, 2022, doi:10.3390/pharmaceutics15010092_

Round 1

Reviewer 1 Report

Introduction:

I would recommend the author slightly reduce the length of the paragraph discussing the EPR effect. While there is a good amount of background information about lectin and PDT, the information regarding the use of magnetite could use more introduction.

In the meantime, I would recommend the author highlight the fact that “IR775 dye” is the API you intend to deliver for anticancer activity. Plus, the lectin and magnetite are serving as targeting molecules.

Results:

I would also recommend adding a description/conclusion after the zeta-potential statement. What do those data suggest?

“Next, we studied the binding of these nanoparticles to cells by fluorescence microscopy. The data presented in Fig. 3c indicate more efficient penetration of mPLGA/IR775-ConA nanoparticles into cells compared to control mPLGA/IR775-BSA nanoparticles.”

I am not sure I understand how mPLGA nanoparticles were uptake by the cells. Is it driven by receptor internalization or clathrin-mediated endocytosis? The author could help the readers understand how is that facilitated.

“Data presented in Fig. 4b demonstrate the difference in nanoparticle accumulation in tumors for these groups: 4.5 ± 1.5% vs. 7.3 ± 1.3 % of the injected dose for groups without and with the applied magnetic field, respectively.”

Please share a detailed description what is the statistical significance between 4.5% vs 7.3%. More specifically, how would such a small difference be translated into larger species of animals (NHPs or humans)? A meaningful discussion would be appreciated.

Author Response

Reviewer #1:

Comment 1

I would recommend the author slightly reduce the length of the paragraph discussing the EPR effect. While there is a good amount of background information about lectin and PDT, the information regarding the use of magnetite could use more introduction.

Reply 1

We thank the reviewer for the valuable comment. We reduced the length of the EPR description and provided a more detailed description of magnetic delivery as follows:

“For example, liposomal forms of chemotherapeutic drugs, such as doxorubicin, penetrate the tumor through the EPR effect (enhanced permeability and retention effect) [6]. However, for the majority of human solid tumors, in contrast to laboratory rodents, the EPR effect is much less pronounced and passive delivery is not as efficient as expected [7].

To solve this problem, different methods of targeted delivery are now being developed, e.g., targeted drug delivery, magnetic delivery, or combination delivery strategies, thus implementing a “magic bullet” concept. Namely, magnetically-enhanced nanoparticle delivery is one of the mainstream directions of modern nanobiotechnology. Magnetite incorporation into nanoparticle structure allows non-invasively visualizing different pathologies within the organism with different MRI regimens [8,9]; quantitatively assessing the nanoparticle accumulations in different tissues in funda-mental applications [10], and performing a magnetically-guided drug delivery using an external source of electromagnetic field near the region of interest, e.g., tumor site [11,12]. Different magnetite-based nanoparticles are already used in clinical applications as MRI contrasting agents thus proving the of this kind of nanoparticles for bio-medicine (Ferucarbotran for hepatocellular carcinoma or Ferumoxide for the imaging of mononuclear phagocyte systems) [13]”.

Comment 2

In the meantime, I would recommend the author highlight the fact that “IR775 dye” is the API you intend to deliver for anticancer activity. Plus, the lectin and magnetite are serving as targeting molecules.

Reply 2

The corresponding paragraph was corrected and now presented as follows:

“Here we demonstrate the rational design of lectin-modified nano-PLGA loaded with PDT sensitizer (IR775) and magnetite, namely, mPLGA/IR775-lectin. The developed nanoparticles realized a combined targeted delivery strategy, namely the combination of active targeting mediated by lectin and magnetically-guided targeting mediated by magnetite loaded inside nanoparticles and external magnetic field. The anti-cancer efficacy is mediated by the loading of photosensitizer, IR775 dye, into PLGA nanoparticle structure. Heptamethine cyanine derivative, IR775 is one of the most ef-fective photosensitizers, especially inside polymer nanoparticles due to its hydrophobic nature, under external light irradiation in the near-infrared window in biological tissue it produces reactive oxygen species (ROS) leading to cancer cell death thus af-fecting only specific tissue site only on demand under light exposure [35–37]”.

Comment 3

I would also recommend adding a description/conclusion after the zeta-potential statement. What do those data suggest?

Reply 3

We thank the reviewer for the valuable comment. Indeed, the conclusion was missed, and now it is added to the main text as follows:

“Electrophoretic light scattering measurements showed nanoparticle ζ-potentials to be equal to -20.2 ± 10.4 mV for mPLGA/IR775, -21.3 ± 4.84 mV for mPL-GA/IR775-BSA, -19.9 ± 6.42 mV for mPLGA/IR775-UEA, -20.3 ± 6.91 mV for mPL-GA/IR775-SBA, -19.2 ± 6.05 mV for mPLGA/IR775-PNA, and -17.7 ± 5.99 mV for mPLGA/IR775-ConA. Since the isoelectric points of conjugated lectins are in the slightly acidic region, pI 4.5 – 5.1 for UEA, pI 5.8 – 6.0 for SBA, pI 5.5 – 6.5 for PNA, and pI 4.5 – 5.5 for ConA, all of the studied lectins are charged neutrally or slightly negatively in buffers with pH 7.0. A possible problem would be that when a significant amount of proteins are conjugated to the nanoparticle surface, the particles can aggregate due to a strong increase in surface charge. However, measurements showed only a slight increase in ζ-potentials due to the optimal concentration of proteins during conjugation, and the particles retained colloidal stability.”.

Comment 4

“Next, we studied the binding of these nanoparticles to cells by fluorescence microscopy. The data presented in Fig. 3c indicate more efficient penetration of mPLGA/IR775-ConA nanoparticles into cells compared to control mPLGA/IR775-BSA nanoparticles.” I am not sure I understand how mPLGA nanoparticles were uptake by the cells. Is it driven by receptor internalization or clathrin-mediated endocytosis? The author could help the readers understand how is that facilitated.

Reply 4

Corrected. The issue is discussed as follows:

“Next, we studied the binding of these nanoparticles to cells by fluorescence microscopy. The data presented in Fig. 4c indicate more efficient penetration of mPL-GA/IR775-ConA nanoparticles into cells compared to control mPLGA/IR775-BSA nanoparticles. Since breast cancer cells overexpress more mannose N-glycan units on the membrane proteins than that of normal cells, the binding of mPLGA/IR775-ConA occurs due to the interaction of ConA with mannose residues on several proteins types in cancer cells. N-glycosylation, the sequential addition of complex sugars to adhesion proteins, ion channels, and receptors, is one of the most frequent protein glycosylation processes, leading to the fact that ConA-based nanoparticles interact both with internalizing and non-internalizing proteins on the cell membrane, resulting in the nano-particle accumulation in different cell compartments – resting on the cell membrane, localizing in endosomes and in the cytoplasm [43–45]. Maybe this feature is one of the most interesting aspects of lectin-modified nanoparticles' interaction with cancer cells gathering the attention of researchers since this fact allows affecting several types of cell compartments simultaneously, which is very important on the road to the development of multifunctional cancer-fighting strategy, especially those based on PDT.”.

Comment 5

“Data presented in Fig. 4b demonstrate the difference in nanoparticle accumulation in tumors for these groups: 4.5 ± 1.5% vs. 7.3 ± 1.3 % of the injected dose for groups without and with the applied magnetic field, respectively.” Please share a detailed description what is the statistical significance between 4.5% vs 7.3%. More specifically, how would such a small difference be translated into larger species of animals (NHPs or humans)? A meaningful discussion would be appreciated.

Reply 5

We really thank the reviewer for a very good question, which really worries us a lot and forces us to develop new tumor models in order to properly answer it. This issue is described in the text as follows:

“Data presented in Fig. 5b demonstrate the difference in nanoparticle accumulation in tumors for these groups: 4.5 ± 1.5% vs. 7.3 ± 1.3 % of the injected dose for groups without and with the applied magnetic field, respectively. Significant differences between the control group and experimental group determined with two-sided unequal variances t-tests (Welch’s test) for two-group comparisons showed the p-value to be equal to 0.0004, thus proving the difference in nanoparticle accumulation in tumors.

Here we showed that the application of a magnetic field increased the delivery of mPLGA/IR775-ConA nanoparticles into the tumor by 1.6 times. Here we should emphasize the problem that always arises on the way to the development and verification of human anti-cancer drugs using laboratory mice. Since mice are small rodents and their tumors are correspondingly small, it is quite difficult to predict how targeted nanoparticles will behave in terms of quantitative accumulation in the big human tumor, especially considering the fact that there are no successful cases of introducing targeted nanoparticles into the clinic and there is no way to obtain relevant data from the literature for at least one of the existing targeted nanoparticles. We can only try to model the processes in rodents that maximally reflect the processes of carcinogenesis in humans. For this purpose, the allograft tumor models were used in this study, namely mouse cancer cells were injected in mice, while the tumors formed for a rather long time period (for almost 2 weeks) and were allowed to grow to a fairly large size, 190 ± 67 mm3 (thus corresponding to approximately 1% from total body weight), developing a normal vasculature, thereby minimizing the EPR effect of fast-growing tumors, like B16 or CT26 models.

However, it is rather presumptuous to suggest that such efficiency and such a difference in the accumulation of nanoparticles with and without a magnetic delivery can be reproduced in human tumors. To shed light on this problem, it is necessary to perform a series of sequential experiments using larger mice, e.g., ICR (CD-1) instead of Balb/c, big rats, and other larger animals, and this is included in our plans for further research”.

Reviewer 2 Report

In this manuscript, Kovalenko et. al., developed PLGA NPs loaded with magnetite and IR755, and conjugated with lectins for cancer targeting. In vitro and in vivo experiments were performed to prove the concept of the study. The concept is interesting; however the data presentation is very poor. So, it should be improved before publication. Some comments are shown below.   

1. The quality of SEM images is very poor.

2. The ROS generation properties of NPs should be performed using DPBF assay or DMA assay.

3. The NPs need to be characterized in terms of UV-Vis absorbance spectroscopy and photoluminescence spectroscopy.

4. What is the loading content of IR755 in NPs?

5. The in vivo tumor volume should be presented as mean±SD as the experiments were performed in replicates?

6. Scale bar is missing in mice fluorescence images.

7. What is the x-axis of Figure 3d?

Author Response

Comment 1

The quality of SEM images is very poor.

Reply 1

Corrected.In order to improve the quality of electron microscopy images, the image sizes were increased to minimize jpeg compression artifacts, new images of polymer nanoparticles were obtained, and TEM images of magnetite cores were obtained, clearly demonstrating their size:

Figure 2. Synthesis and characterization of mPLGA/IR775 nanoparticles. (a) Scanning electron microscopy of magnetite synthesized with the coprecipitation technique. The inset is transmitting electron microscopy of magnetite. (b) Scanning electron microscopy of PLGA nanoparticle loaded with magnetite and IR775 photosensitizer. The SEM inset demonstrates the incorporation of magnetite – red arrows show white dots corresponding to magnetite cores. (c) Physical size distribution of magnetite obtained via SEM image processing. (d) Physical size distribution of PLGA nanoparticles loaded with magnetite and IR775 (mPLGA/IR775). (e) UV-Vis absorbance spectrum of mPLGA/IR775 and PLGA/IR775 nanoparticles. (f) Excitation and emission fluorescence spectra of mPLGA/IR775 nanoparticles: blue – fluorescence excitation (measured with λem = 800 nm), fluorescence emission (measured with λex = 700 nm).

Comment 2

The ROS generation properties of NPs should be performed using DPBF assay or DMA assay.

Reply 2

Since DPBF and DMA sensors have a fluorescence excitation peak very close to the first excitation peak of IR775 (about 440 nm), in order to prevent the possible interference of these substances, the ROS generation was measured using the equal sensor CM-H2DCFDA (with λex = 500 nm) as follows:

“The effective reactive oxygen species (ROS) production of PLGA/IR775 and mPLGA/IR775 under light irradiation was evaluated using a CM-H2DCFDA sensor pre-mixed with nanoparticles. The mixture was light-irradiated, particles were centrifuged and the fluorescence of the supernatant was measured (Fig. 3a). Data presented in Fig. 3a confirm that particles produced ROS under light irradiation. Interestingly, the ROS production is 2.5 higher for nanoparticles loaded with magnetite and IR775 in comparison with particles loaded with IR775 only under light irradiation (Fig. 3a).

Figure 3. Characterization of mPLGA/IR775 nanoparticles as nanoagents for PDT. (a) ROS generation study of nanoparticles under the irradiation with 808 nm laser: magnetic mPLGA/IR775 and non-magnetic PLGA/IR775 nanoparticles were incubated with ROS sensor and irradiated with 808 nm light, centrifuged and the fluorescence of supernatant was measured (λex = 500 nm, λem = 525 nm). (b) Flow cytometry study of ROS generation. Cells were incubated with mPLGA/IR775 nanoparticles and ROS sensor and irradiated with 808 nm laser for 10 min. Histograms represent cell populations in the fluorescent channel corresponding to the ROS sensor fluorescence (λex = 488 nm, λem = 525/20 nm). (c) mPLGA/IR775 were conjugated to different lectins as well to BSA and characterized with dynamic light scattering method confirming the colloidal stability of nanoparticles after chemical conjugation”.

Comment 3

The NPs need to be characterized in terms of UV-Vis absorbance spectroscopy and photoluminescence spectroscopy.

Reply 3

UV-Vis absorbance of nanoparticles as well as the fluorescence excitation and emission was measured and presented in Fig. 2e and Fig. 2f, respectively.

Comment 4

What is the loading content of IR755 in NPs?

Reply 4

The loading content of IR775 was measured, the procedure described in the Methods section, and this experiment is presented in the main text as follows:

“The measurement of IRR75 loading efficiency showed that the dye loading is equal to 16.8 ± 0.8 µg of IR775 per 1 mg of mPLGA/IR775, thus demonstrating 80.6 % loading efficiency during the synthesis (250 µg of IR775 were used in synthesis per 12 mg of PLGA)”.

Comment 5

The in vivo tumor volume should be presented as mean±SD as the experiments were performed in replicates?

Reply 5

Corrected. All tumor volumes are presented as mean ± SD throughout the manuscript.

Comment 6

Scale bar is missing in mice fluorescence images.

Reply 6

The scale bar was added to the fluorescence images (Fig. 5a and Fig. 6a).

Comment 7

What is the x-axis of Figure 3d?

Reply 5

The x-axis is the time of laser irradiation; the axis label is added to the figure

Round 2

Reviewer 2 Report

The authors revised the manuscript according to the comments provided by the reviewer. So, it can be accepted now for publication.